# Woody Above-Ground Biomass Estimation on Abandoned Agriculture Land Using Sentinel-1 and Sentinel-2 Data

**Tomáš Bucha** [1,*] **, Juraj Papčo** [2] **, Ivan Sačkov** [1] **, Jozef Pajtík** [1] **, Maroš Sedliak** [1] **, Ivan Barka** [1] **and Ján Feranec** [3]

1   National Forest Centre—Forest Research Institute, T. G. Masaryka 22, 960 01 Zvolen, Slovakia; sackov@nlcsk.org (I.S.); pajtik@nlcsk.org (J.P.); sedliak@nlcsk.org (M.S.); barka@nlcsk.org (I.B.)
2   Department of Theoretical Geodesy and Geoinformatics, Faculty of Civil Engineering, Slovak University of Technology in Bratislava, Radlinského 11, 810 05 Bratislava, Slovakia; juraj.papco@stuba.sk
3   Institute of Geography, Slovak Academy of Sciences, Štefánikova 49, 814 73 Bratislava, Slovakia; feranec@savba.sk
*   Correspondence: bucha@nlcsk.org

**Abstract:** Abandoned agricultural land (AAL) is a European problem and phenomenon when agricultural land is gradually overgrown with shrubs and forest. This wood biomass has not yet been systematically inventoried. The aim of this study was to experimentally prove and validate the concept of the satellite-based estimation of woody above-ground biomass (AGB) on AAL in the Western Carpathian region. The analysis is based on Sentinel-1 and -2 satellite data, supported by field research and airborne laser scanning. An improved AGB estimate was achieved using radar and optical multi-temporal data and polarimetric coherence by creating integrated predictive models by multiple regression. Abandonment is represented by two basic AAL classes identified according to overgrowth by shrub formations (AAL1) and tree formations (AAL2). First, an allometric model for AAL1 estimation was derived based on empirical material obtained from blackthorn stands. AAL2 biomass was quantified by different procedures related to (1) mature trees, (2) stumps and (3) young trees. Then, three satellite-based predictive mathematical models for AGB were developed. The best model reached $R^2 = 0.84$ and RMSE = 41.2 t·ha$^{-1}$ (35.1%), parametrized for an AGB range of 4 to 350 t·ha$^{-1}$. In addition to 3214 hectares of forest land, we identified 992 hectares of shrub–tree formations on AAL with significantly lower wood AGB than on forest land and with simple shrub composition.

**Keywords:** farmland overgrowth; shrub–tree formations; biomass estimation; satellite data; radar backscatter; coherence; regression model; integrative management

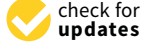



## 1. Introduction

Between 1990 and 2015, the world lost 1.29 million square kilometres of forest [1]. Contrary to the global trend, forest area in Europe expanded by 0.17 million km$^2$ between 1990 and 2010 [2]. The expansion of European forests was the result of afforestation (planting and seeding of trees on land that was not previously forested) and the natural expansion of forests, such as on abandoned land.

The phenomenon of the abandonment of the agricultural landscape is especially notable in the countries of Eastern and Central Europe, where formerly intensively worked farmland was abandoned due to deep social and political changes such as the disintegration of socialist agrarian policies, the accession to the EU and countries joining the global markets [3,4]. An example of such agricultural land changes is in Slovakia, where unused areas amount to 424–452 thousand hectares (ha), representing ~18% of the country's farmland. The use of these areas does not correspond to the records of land categories at the Office of Geodesy, Cartography and Cadastre. The process of overgrowth with shrubs and trees is largely associated with unused "permanent grasslands" and represents up to ~336 thousand hectares [5].

After years of disinterest, areas of abandoned agricultural land (AAL) have become the subject of societal debate due to the extraction of wood intended for combustion in bio-energy facilities. This extraction results from broader strategic decisions. The EU 2030 climate and energy framework set a target to increase the share of energy production from renewable sources to 32%. Moreover, continuous forest and shrub growth outside of actively managed forests must be included in the carbon storage accounting (EU Regulation 2018/841).

The most common approaches applied to identify AAL in the Central European region are based on comparing land cover from two or more time periods. Using satellite images or existing maps, the comparison is based on the changes in spectral and textural properties or forest boundaries from long-term forest mappings [6–8]. The height of vegetation derived from aerial orthophotos or lidar data has been used as an indicator of AAL and its timing [9,10]. A review paper [11] analysed studies published from 1992 to 2019 to identify agricultural land abandonment by applying satellite optical and microwave data. The authors concluded that the studies mostly did not consider detailed field surveys, and an assessment of natural vegetation overgrowth in AAL was also missing. Only a few studies specifically addressed the issue of estimating woody above-ground biomass (AGB) on AAL in Europe from satellite images, such as [12] from the point of view of carbon sequestration. The authors in [13] emphasised the role of new technologies such as Google Earth Engine (GEE) that makes much of the free satellite imagery available online so that researchers can analyse changes to the Earth's surface in near real-time and can run algorithms on a large archive of georeferenced images.

As the wood biomass outside of forest land has not been precisely inventoried [11], only limited information is usually available on the national level from surveys such as the National Forest Inventory (NFI). There are several problems here: (i) deriving reliable data at levels lower than the national level poses a problem because of the applied sampling rate; (ii) due to cost, repeat surveys are carried out mostly at 10-year intervals; (iii) allometric models that allow the prediction of AGB have not yet been developed for shrub species; and (iv) studies on woody biomass are mainly related to economically important forest trees species. Although there are models for successive trees with characteristic occurrence on AAL (e.g., birch, aspen, pine, rowan, goat willow), these refer to trees growing in continuous canopies. On AAL, these species occur mainly as solitaires. Without precise information on the AAL area, structure, volume and increment, it is impossible to set up a sustainable management system. Such a situation is unsatisfactory in terms of biodiversity and landscape aesthetics as well.

Due to the extensive overgrowth of agricultural land [2], satellite RS is an effective way of identifying and quantifying AGB on AAL. The relationship between the SAR backscatter and the AGB shows that the reflection intensity increases with increasing biomass until it reaches a saturation level. Due to issues with backscatter saturation of shorter wavelength X- and C-bands [14], relatively longer wavelength SAR systems such as L- and P-bands have been promoted for characterizing AGB in dense or intact forested landscapes for many years [15,16]. A comparison of Sentinel-1 and PALSAR-2 data showed a Sentinel C-band saturation at around 50 t·ha$^{-1}$, while the L-band PALSAR-2 had a saturation point around 150 t·ha$^{-1}$ at VH polarization [17]. Similarly, it was concluded in [16] that current space-borne sensors (radar and optical) are inadequate for accurately estimating AGB beyond 100–150 t·ha$^{-1}$.

Besides saturation, another problem is the variation in the radar backscatter, mainly during freeze–thaw cycles accompanied by large variations in temporal coherence [18]. Most of the studies tried to solve the problem of radar signal saturation and temporal data variation to achieve the highest accuracy in quantifying the amount of biomass by integrating multi-frequency SAR data from different sensors [19], combining radar and optical satellite or airborne data [20,21], using multi-temporal data [17,22] or using new or improved algorithms for biomass retrieval. A review of SAR techniques and methods showed several approaches using both parametric and nonparametric methods

to estimate AGB, further grouped into data-driven and model-driven methods [16,20]. Biomass estimation remains challenging, especially in areas with complex shrub and tree stand structures and diverse forests and environmental conditions [23].

The primary goal of this study was to propose, experimentally prove and validate the concept of satellite-based AGB estimation on AAL in the Western Carpathian region.

Therefore, the first objective was to propose a method for AAL identification, with a preference for using publicly available and free data. In the study, we verified a combination of Sentinel-2 satellite images with data from the Land Parcel Identification System and the National Cadastre.

The second objective was to derive an allometric model for shrub AGB estimation on AAL. Due to the lack of such models, we carried out a precise quantification of shrub AGB biomass on AAL sample plots in line with the recommendations for reducing uncertainty in AGB enumeration [20].

From an RS point of view, the problem of nonlinearity and satellite signal saturation with increasing AGB and the occurrence of complex structures formed by shrub and tree formations can be eliminated by combining Sentinel-1 (C-band) and Sentinel-2 data. The preference for Sentinel data follows from two assumptions: (i) their free availability creates a condition for the operational deployment of the proposed concept of AGB estimation; (ii) as the biomass on AAL is expected to be markedly lower than in forest stands, the problem of saturation could be solved e.g., by the selection of suitable bands and period of acquisition. Therefore, the third objective was to separately analyse Sentinel-1 and -2 data to find the optimal band combinations, testing the backscattered amplitude and the interferometric coherence and their seasonal variation to determine the optimal period for AGB retrieval on AAL.

The fourth objective was to verify whether AGB estimation could be improved by integrating our individual results with promising partial conclusions from previous studies [17,21,22] which have not yet been tested in synergy. This meant creating and testing integrated AGB regression models that combined radar and optical satellite data, multitemporal data, polarimetric coherence and backscatter. The decision to apply empirical multiple regression models for AGB estimation came from their simplicity, comprehensibility and good performance, which are important for operational deployment. At present, the application of advanced techniques for AGB estimation was particularly limited by an insufficient amount of in situ data necessary for their use.

## 2. Materials and Methods

### 2.1. Study Area

The study area is situated in the Western Carpathians in central Slovakia in the geomorphological units of the Zvolenská kotlina basin and the Javorie Mountains (Figure 1a) with centre coordinates of 48°32′N, 19°21′E. The region belongs to the European temperate climate zone. The experimental territory area was 12,518 hectares and corresponded to the boundaries of the Viglas forest management unit (FMU) (Figure 1b). Agricultural land covers 8477 hectares, and forest land covers 3214 hectares. The altitude of the area ranges from 320 to 944 metres a.s.l. The agricultural exploitation of the territory is described in detail in our previous study [10].

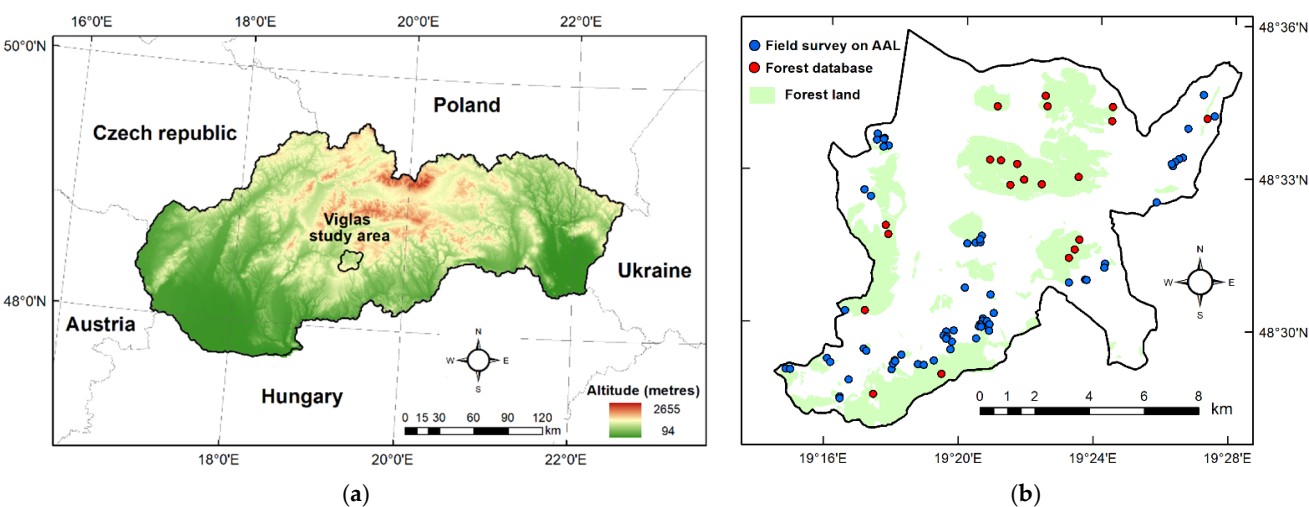

(**a**)                                   (**b**)

**Figure 1.** (**a**) Location of the Viglas study area in central Slovakia (black polygon); (**b**) localization of the field survey plots on abandoned agricultural land (blue circles) and the plots from forestry database (red circles) in study area.

### 2.2. Concept of Biomass Estimation on Abandoned Agricultural Land

The estimation of wood biomass on AAL was based on freely available data from satellite archives (Sentinel hub), the National Cadastre and the Land Parcel Identification System (LPIS). Data availability was essential for the operational deployment of the methodology. Empirical modelling using Sentinel-1 radar backscatter and coherence and Sentinel-2 DN values was applied to predict wood AGB on AAL. Airborne laser scanning and digital aerial images were used for sampling design and for precisely identifying and distinguishing the sampling plots. Figure 2 depicts a simplified flowchart of the concept.

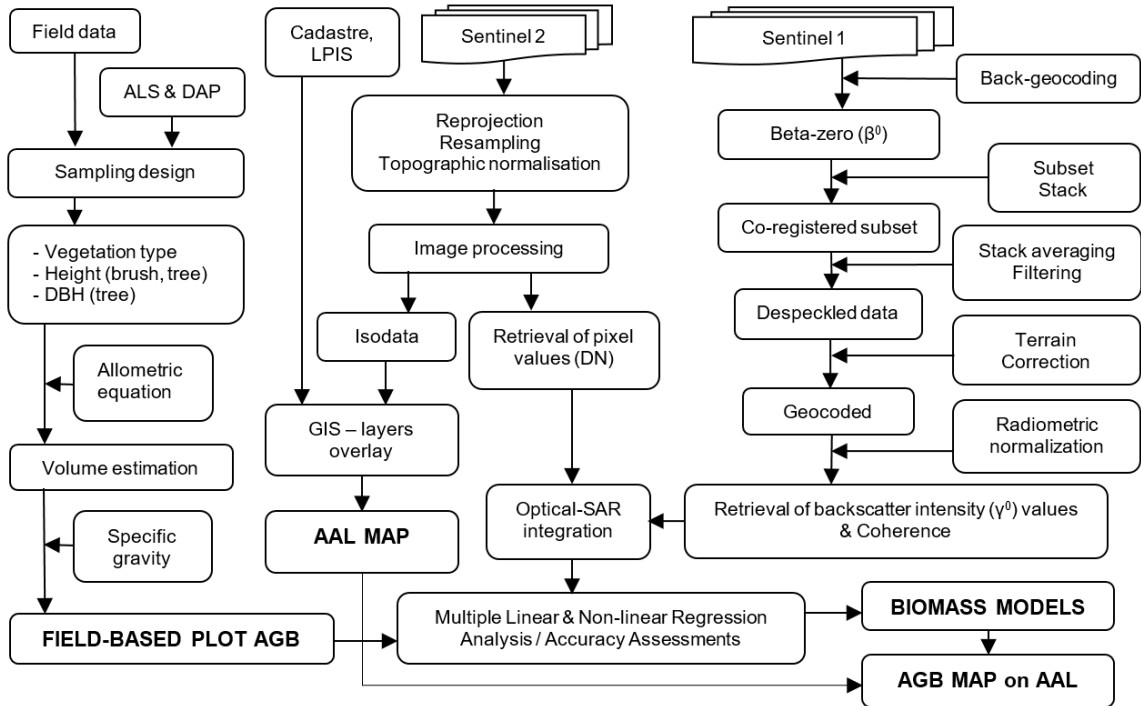

**Figure 2.** Flowchart of the spatial identification and estimation of AGB on AAL. LPIS, Land Parcel Identification System; ALS, airborne laser scanning; DAP, digital aerial photography.

The study was based on a general definition of AAL: land devoid of any activities associated with agricultural production until it becomes overgrown by vegetation other

than agricultural crops [11]. For the purpose of selecting training plots within field surveys, two basic AAL classes were identified according to the physiognomy of overgrowth by various species of wood vegetation, and their height, density and clustering:

AAL1: Abandoned agricultural land overgrown by medium-sized vegetation (shrub formations): originally agricultural land (arable land, meadows and pastures, vineyards and orchards) fully overgrown by grasses and broad-leaved herbs and shrubs with canopy closure of >20% and with a maximum height of 3 m.

AAL2: Abandoned agricultural land overgrown by tall vegetation (tree formations): originally agricultural land (arable land, meadows and pastures) fully overgrown by grasses and broad-leaved herbs and shrubs with a varied canopy closure and with >20% tree canopies taller than 3 m.

### 2.3. AAL Identification

In spatially identifying AAL in the study area without distinguishing classes, we used a straightforward approach based on a combination of available cadastral data, LPIS and Sentinel-2 imagery.

The National Cadastre is a public land registry and information system containing, inter alia, geometric determinations and data about the types of land parcels. Cadastral parcels coded as arable land, permanent grassland (meadows and pastures) and orchards in the GIS vector format from 2018 were used in the study.

The Land Parcel Identification System (LPIS) is an integrated administrative and control information system based on photographs of agricultural parcels used to check payments made under the EU Common Agricultural Policy (CAP). The LPIS contains a vector layer of agricultural parcels and information about crop and land use, which are used on farmers' applications for subsidies. We assumed that such parcels represented actively managed agricultural land. Data in the shapefile vector format from 2018 were used in the study.

The LPIS vector boundaries of actively managed agricultural land (arable land, meadows and pastures) and cadastral parcels registered as agricultural land (of the same type) were overlaid to identify the land considered by LPIS as uncultivated. Those cadastral parcels that fell outside the LPIS blocks were considered as potentially abandoned agricultural land. Then we filtered out pixels under snow cover using Sentinel-2 imagery from the winter season; i.e., we masked arable land, meadows and pastures. Thus the remaining areas represented real abandoned agricultural land with woody vegetation. For accurate assessments, we compared the classifications with a reference dataset on 127 points randomly selected in areas identified as potentially AAL. As a reference dataset, we used airborne CIR images.

### 2.4. Field Survey

Field data on AAL were collected during field surveys in the study area in 2018 and 2019. In all, 56 plots ranging in their biomass from 4 to 350 t·ha$^{-1}$ were selected on AAL to represent the height range and heterogeneity of shrub and shrub–tree formations. Square plots were established with sides varying from 10 to 30 m according to the shrub–tree stand density. There were 30 pure shrub plots, 10 mixed plots with shrubs and trees and 16 pure tree plots. To ensure a more even representation of the plots in the whole range of AGB, the database was extended by tree plots from a stand-wise forest inventory (available from the National Forestry Database). A total of 21 plots were selected by random sampling, ranging in their biomass from 100 to 350 t·ha$^{-1}$. The size varied from 1.02 to 21.6 hectares, with a mean of 4.58 hectares (see Table 1). In total, the reference database was comprised of 77 plots (Figure 1b). In the first group, 56 shrub–tree plots were predominantly covered by blackthorn (*Prunus spinosa* L.) and dog rose (*Rosa canina* L.). In the second group, 21 tree plots were predominantly covered by broad-leaved species such as European beech (*Fagus sylvatica* L.) and European hornbeam (*Carpinus betulus* L.). Other tree species, such as black locust (*Robinia pseudoacacia* L.), small-leaved lime (*Tilia cordata* Mill.), sessile oak (*Quercus*

*petraea*, Matt.), field maple (*Acer campestre*, L.), European ash (*Fraxinus excelsior*, L.), Norway spruce (*Picea abies* L.) and wild cherry (*Prunus avium* L.) occurred individually or in small groups in some of the plots in both groups.

**Table 1.** Reference database: descriptive statistics of ground plots.

| | | | AGB | | |
|---|---|---|---|---|---|
| | Number of Plot | Mean Size (ha) | Mean (t·ha$^{-1}$) | Min–Max (t·ha$^{-1}$) | Percentiles 25–75% (t·ha$^{-1}$) |
| All plots | 77 | 1.36 | 117.5 | 4.4–336.6 | 26–210 |
| Shrub-tree plots on AAL | 56 | 0.16 | 76.9 | 4.4–336.6 | 22–99 |
| Tree plots on FL | 21 | 4.58 | 225.5 | 110.3–328.9 | 166–281 |

AAL, abandoned agriculture land; FL, forest land.

### 2.4.1. Shrub Biomass Estimation

As regional biomass tables are not available for shrub vegetation, the allometric model for calculating the dry matter of the above-ground biomass of shrubs was derived based on the empirical material obtained from blackthorn stands in the study area, which made up about 90% of all the bushes.

All the above-ground shrub biomass was cut out from 20 pure shrub plots and weighed on a sub-area of 2 × 2 m (Figure 3a). The mean height was calculated from five heights measured in the corners and middle of each 2 × 2 m sub-area. The fresh weight of the removed biomass was determined using a hanging scale. The sub-samples were then dried to a constant weight in an oven at 105 °C, and the dry matter content of the wet samples was determined. An allometric model was derived for the calculation of AGB of blackthorn (*Prunus spinosa* L.), which was the most abundant shrubby species in the model areas. This model was used to calculate the biomass of other, less represented shrub species.

The model was based on one independent variable (mean height) and was expressed as the dry matter of the above-ground biomass per 1 m$^2$ at full canopy. The following allometric equation was used to calculate the biomass:

$$m_{AGB} = b_0 h^{b1} \tag{1}$$

where $m$ is the weight of AGB in kilograms per m$^2$, $h$ is mean height in metres, and $b_0$ and $b_1$ are regression coefficients.

Biomass ($m_{AGB\text{-}SC}$ in kg) over known areas ($S$ in m$^2$) and canopy closure ($C$ in %) was calculated as:

$$m_{AGB-SC} = b_0 h^{b1} \cdot S \cdot \frac{C}{100} \tag{2}$$

Canopy closure on all the AAL plots was estimated with rounding to 5%. The model accuracy of 23.9% (Table 2) is slightly lower than in similar models, where it is around 10–20% [24].

**Table 2.** Allometric model for quantifying above-ground woody biomass of shrubs on abandoned agricultural land.

| Vegetation Formation | Model | n | R$^2$ | RMSE (%) | *p*-Value |
|---|---|---|---|---|---|
| Shrubs (blackthorn) | $m_{AGB} = 1.2417 \times h^{1.45361}$ | 20 | 0.81 | 23.9 | <0.001 |

### 2.4.2. Tree Biomass Estimation

Tree biomass was quantified by 3 procedures related to (1) mature trees, (2) stumps and (3) young trees:

1. Mature tree volume was determined according to Czech–Slovak volume tables [25]. This empirical material includes 18,087 sample trees from areas across Slovakia and Czechia. The model predictors are tree height and diameter at breast height (DBH) for

selected tree species. The volume tables contain volume equations for 11 economically important tree species and 4 volume units (stem, over 7 cm thick, over 3 cm thick, whole tree) with or without bark. We used the volume of the whole tree with the bark ($m^3$). This unit represents the volume without the stump.

2.  Stump volume was calculated according to [26] using the following formulas, for broadleaf and coniferous trees, respectively:

$$V_p = 0.465337 \times \left(1.0376 \times h_p^{-0.0274} \cdot d_{1.3}\right)^{2.094175} \cdot h_p^{1.060645} \tag{3}$$

$$V_p = 0.724703 \times \left(1.0376 \times h_p^{-0.0274} \cdot d_{1.3}\right)^{2.014485} \cdot h_p^{1.026424} \tag{4}$$

where $V_p$ is the stump volume ($m^3$ with the bark), $d_{1.3}$ is the breast diameter (cm), and $h_p$ is the stump height (m), and a default value of 0.3 m was used in the calculations.

3.  The biomass models for young trees up to 10 m were taken from [27]. The models calculate the dry above-ground biomass of individual trees based on tree height and thickness at the base of the trunk for 11 tree species.

The growing stock volume (GSV in $m^3$) of mature trees on the plot was determined by adding up the volume of all the individual trees and stumps. Dividing the GSV by the plot area, we obtained the GSV per unit area ($m^3 \cdot ha^{-1}$). When determining the dry AGB in $t \cdot ha^{-1}$, we made a conversion based on the specific weight (bulk density) taken from [28].

In the case of young trees, we summed the AGB of all individual trees and divided the result with the plot area to calculate the AGB per unit area in $t \cdot ha^{-1}$.

In the case of the co-occurrence of shrubs and young or mature trees, the total AGB on the plot in $t \cdot ha^{-1}$ was calculated as the sum of their AGB on the plot per hectare. Note that assimilatory organs were not included in the calculations.

### 2.4.3. Shrub-Tree Ground Plots Extension

The derivation of statistical characteristics for 56 square shrub-tree plots 100 to 900 $m^2$ in size could be negatively influenced by speckle noise, mainly in the radar data. Therefore, a further step involved preparing a vector layer from the field survey data on all AAL plots. This included (i) defining the homogeneous areas around the plots and (ii) assigning the database of measured data (AGB and woody composition) to the vector layers:

1.  The homogeneous areas around the plots were derived using aerial images (Figure 3b) and a normalised digital surface model (nDSM) layer (Figure 3c). Around each plot, an area with homogeneous vegetation cover was designed by a human operator with experience in GIS and remote sensing. The size of the identified homogeneous areas varied from 0.05 to 0.52 hectare, with a mean of 0.16 hectare (see Table 1).

2.  Measured and calculated data on AGB per hectare of each plot were stored in the database and joined to the field plot vector layer created in step 1. The result was a spatially georeferenced vector layer with attributes of the AGB and the woody species composition, which enabled the next step to be performed: the extraction of the plot's statistical characteristics from the satellite data using zonal statistics.

### 2.5. Satellite Data

Satellite data from Sentinel-1 and Sentinel-2 mission acquired from the Copernicus Open Access Hub were utilized in this study (Table 3).

**Table 3.** Sentinel-1 and Sentinel-2 data and derived products used in the study.

| Sensor/Product | Bands/Predictors | Remark |
|---|---|---|
| Sentinel-1/Level-1 SLC | VV, VH | 60 images from ascending pass (track 175) and 60 images from descending pass (track 51): 1 September 2017 to 30 September 2018 |
| Sentinel-1/ stack average | $\gamma^\circ_{VH}$ $\gamma^\circ_{VV}$ | Whole sample for ascending and descending pass: 1 September 2017 to 30 September 2018 |
| | | Stratum 1: Leaf-on period, 1 September to 13 October 2017 and 21 April to 30 September 2018 |
| | | Stratum 2: Leaf-off period with snow cover, 6 December 2017 to 22 March 2018 |
| | | Stratum 3: Leaf-off period without snow cover, 17 October to 30 November 2017 and 30 March to 17 April 2018 |
| Sentinel-1/coherence | $Coh_{VH}$ $Coh_{VV}$ | 26 coherence image pairs in 6-day steps based on a combination of S1A and S1B acquisitions |
| Sentinel-2/S2A | B4, B5, B8, B11 | 4 images: Leaf-off season with snow, 28 January 2017; leaf-off season without snow, 29 March 2017; top of vegetation season, 22 June 2016; end of vegetation season, 30 September 2018 |

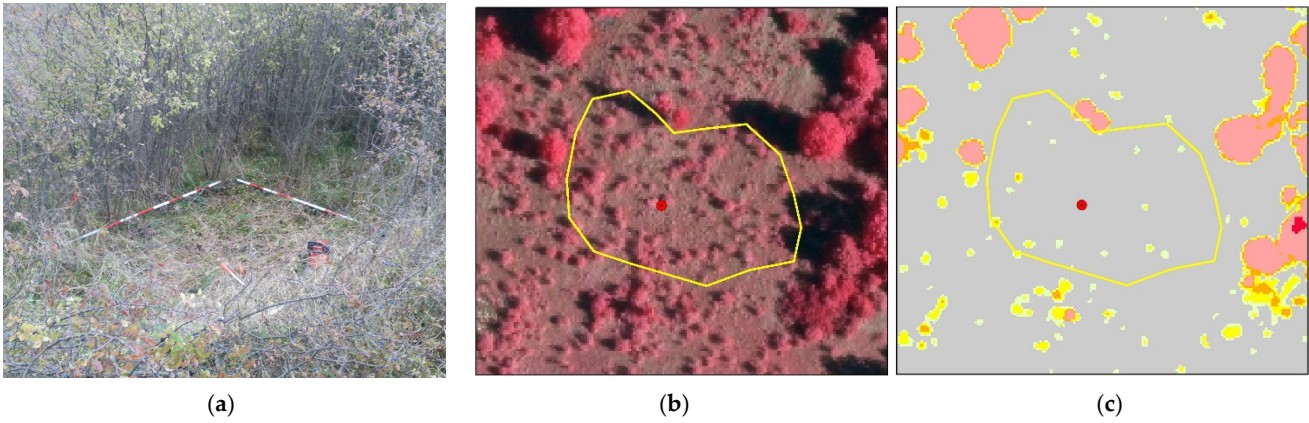

(**a**)          (**b**)          (**c**)

**Figure 3.** (**a**) Field survey: cut out of biomass from a sub-area of 2 × 2 m; (**b**) the identification of homogeneous land cover areas around the field survey plots based on aerial CIR images; (**c**) nDSM layer derived for airborne Lidar survey.

Sentinel-1 interferometric wide swath SLC products were acquired from 1 to 30 September 2017. The data were co-registered to master images selected at the beginning of the vegetation period on 3 April 2018 (DSC pass) and 17 April 2018 (ASC pass). TOPSAR-split was applied to reduce the bursts for selected the AOI. Then precise orbit files were applied, followed by calibration for beta-zero ($\beta^\circ$) creation as prime backscatter information. For relative co-registration to one selected master the back-geocoding approach was used, with a Shuttle Radar Topography Mission (SRTM) 1 arc-second digital elevation model (DEM) and bisinc 5-point interpolation. The next step was to create two stacks of images for ascending and descending passes and averaged products for polarization VV and VH. The last steps involved performing earth and terrain correction using the SRTM 1 arc-second DEM, speckle filtering using a refined Lee filter, and creating gamma-zero ($\gamma^\circ$) using band mathematics ($\gamma^\circ = \beta^\circ/\tan\theta$, where $\beta^\circ$ is the radar backscattering coefficient and $\tan\theta$ is the projected local incidence angle).

The pre-processed database was divided into 3 groups (leaf-on period and leaf-off periods with and without snow cover) to examine the influence of the seasonal variability of the radar backscatter on the accuracy of biomass estimation (Table 3). Interferometric coherence ($\rho$) was estimated from co-registered complex images using a window size of 10 in range and 3 in azimuth direction.

Sentinel-2 satellite data (processing Level-2A, bottom of atmospheric reflectance) were acquired from 4 periods (Table 3), representing the peak vegetation season, the leaf-off vegetation season with snow, the leaf-off vegetation season without snow and the autumn season. Bands from the red, near and short-wavelength infrared spectra were included in the analysis: B4 (664.6 nm), B5 (704.1 nm), B8 (832.8 nm) and B11 (1613.7 nm of central wavelength), with a 10 and 20 m resolution for all 4 periods. The digital number (DN) values, which represent reflectance multiplied by 10,000, were analysed.

All products were transformed into the WGS1984 UTM Zone 34N cartographic reference system with a final pixel resolution of $10 \times 10$ m.

### 2.6. Statistical Models for AGB Estimation

The mean of the topographically corrected backscatter $\gamma^\circ$, the mean of coherence $\rho$ (Sentinel-1 HV, VV) and the mean of DN (Sentinel-2 bands 4, 5, 8, 11) were extracted for individual plots, including the homogeneous areas defined around the plots. GIS zonal statistics were used for the mean calculation. Thus, modelling was based on per-object information. Derived prediction models were applied at the pixel level, i.e., the AGB was estimated for each pixel. We used this approach because, unlike forest stands, wood biomass on AAL is not mapped and therefore, there are no spatial units to which the derived model could be applied.

We defined 3 statistical models—MR1, MR2 and MPW—to link the AGB with the satellite observables. MR1 represents a multiple linear regression model where stepwise backward regression was applied to find an optimal band combination to estimate the AGB:

$$y_i = \beta_0 + \beta_1 x_{1i} + \beta_2 x_{2i} + \beta_3 x_{3i} + \ldots + \beta_n x_{ni} + \varepsilon_i \tag{5}$$

The MR2 model is an extension of MR1 that includes additional terms. It is expected that by including them in the AGB model, the model performance will increase. We tested additional terms that account for the interaction between any two predictor variables i.e., one variable divided or multiplied by another. Using $R^2$ we selected terms that were most beneficial to the model's explanatory power.

The third model expresses the dependence between the AGB and the predictor variables using a multiple power function (MPW) of the form:

$$y_i = \beta_0 \cdot x_{1i}^{\beta_1} \cdot x_{2i}^{\beta_2} \cdot x_{3i}^{\beta_3} \cdot \ldots \cdot x_{ni}^{\beta_n} \cdot \theta \tag{6}$$

where $y_i$ is the AGB to be predicted for plot $i$, $x_{1i}$–$x_{ni}$ are independent variables ($\gamma^0$, $\rho$ or DN plot mean), $\beta_1$–$\beta_n$ are coefficients of the model, $\varepsilon_i$ is the model's error term (residuals), and $\theta$ is the multiplicative error.

Logarithmic transformation of the power Equation (6) was used to remove heteroscedasticity and achieve a normal residue distribution. The relationship has the following form:

$$\ln y_i = \beta_0 + \beta_1 \ln x_{1i} + \beta_2 \ln x_{2i} + \beta_3 \ln x_{3i} + \ldots + \beta_n \ln x_{ni} + \varepsilon_i \tag{7}$$

The logarithmic transformation caused a bias. This was corrected by the correction factor $\lambda$ using the method introduced in [29] and applied in our previous study [27]. As a result of interpretation of the linearized model, the original scale (t·ha$^{-1}$) requires the retransformation of Equation (7):

$$y_i = e^{(\beta_0 + \beta_1 \ln x_{1i} + \beta_2 \ln x_{2i} + \beta_3 \ln x_{3i} + \ldots + \beta_n \ln x_{ni})} \lambda \tag{8}$$

The regression function, the Pearson correlation, the coefficient of determination ($R^2$) and the F-test of statistical significance of the regression models (*p*-value) were calculated to assess the strength of the relationship between the AGB and the predictor variables. These were included in the model if they were statistically significant at $p < 0.05$. The best model among the regression methods (MR1, MR2, MPW) was selected based on the adjusted $R^2$.

### 2.7. Validation of AGB Estimation

The models' accuracy was assessed using a bootstrap method with 500 repetitions [30]. Measures of the models' accuracy were obtained using the coefficient of determination ($R^2$), the root mean square error (RMSE) and the bias. The obtained performance parameters were subsequently averaged over the 500 bootstrap iterations. The averaged RMSE and the bias were then divided by the mean AGB value and multiplied by 100 to create RMSE% and bias%.

In a detailed analysis of the models, we applied the approach used by the authors in [16]: a set of reference AGB values, $B_{ref}^{(i)}$, and their estimates, $B_{est}^{(i)}$, where the reference are restricted to a given range $B_1 \leq B_{ref}^{(i)} < B_2$ (0–100, 100–200 and 200–350 t·ha$^{-1}$). For each range, we calculated *bias b* as the average value of the error $B_{est}^{(i)} - B_{ref}^{(i)}$ and the standard deviation of the error $\sigma$. The RMSE in the specified range is given by $\sqrt{\sigma^2 + b^2}$, and the relative RMSE as $RMSE/\overline{B}_{ref} \times 100$. The coefficient of variation (CV) of the error is given as $\sigma/b$. When CV exceeds 1, RMSE is dominated by a random error; when it is less than 1, the bias is the dominant error source in the estimator.

All relationships were investigated for the backscatter coefficient, coherence, and DN value as independent variables (predictors) and the AGB as a dependent variable (estimates). All the empirical models were evaluated based on the combination of strength, direction and significance, yielding a categorization of the model's accuracy in 6 categories ranging from highly to marginally significant, with positive and negative slopes of the regression line (see the explanation below the Table 4).

**Table 4.** Correlation coefficient between AGB and S1 $\gamma^\circ$ radar backscatter in descending order.

| S1 $\gamma^\circ$VH | r | S1 $\gamma^\circ$VV | r |
|---|---|---|---|
| leaf-off (des, s3) | 0.79 +++ | leaf-off (des, s3) | 0.77 +++ |
| leaf-off (asc_des, s3) | 0.77 +++ | leaf-off (asc_des, s3) | 0.77 +++ |
| leaf-off-snow (des, s2) | 0.76 +++ | leaf-off-snow (des, s2) | 0.75 +++ |
| leaf-off-snow (asc_des, s2) | 0.72 +++ | leaf-off-snow (asc_dec, s2) | 0.73 +++ |
| leaf-off (asc, s3) | 0.64 +++ | leaf-on (des, s1) | 0.66 +++ |
| leaf-on (des, s1) | 0.57 +++ | leaf-on (asc, s1) | 0.60 +++ |
| leaf-off-snow (asc, s2) | 0.57 +++ | leaf-off (asc, s3) | 0.60 +++ |
| leaf-on (asc, s1) | 0.56 +++ | leaf-off-snow (asc, s2) | 0.54 +++ |
| whole sample | 0.57 +++ | | 0.66 +++ |

asc, ascending mode; des, descending mode; asc_des, average of ascending and descending images. s1: stratum 1, leaf-on period; s2: stratum 2, leaf-off period with snow; s3: stratum 3, leaf-off period without snow. +++ highly significantly positive ($p < 0.001$).

## 3. Results

### 3.1. Predictor Variable Pre-Selection

The relationships between the AGB and the evaluated predictor variables, expressed by the Pearson correlation coefficient values, are shown in Tables 4 and 5 and Figure 4. The predictor variables are clustered into three groups: Sentinel-1 radar backscatter, Sentinel-1 coherence data and Sentinel-2 DN.

In the Sentinel-1 radar backscatter, the averages from the multi-temporal images were calculated for each stratum to reduce the speckle in the single images and to investigate the impact of the backscatter variation on the AGB estimation. The average $\gamma^\circ$VH and $\gamma^\circ$VV from a leaf-off period without snow cover were found to have the highest correlation with the AGB ($r_{VH} = 0.79$ and $r_{VV} = 0.77$). Although we achieved the highest correlation between the AGB and $\gamma^\circ$ at the descending passes, we included in the prediction model a variable calculated as the average of the ascending and descending passes. Averaging images from both tracks reduces the problems of artefacts caused by foreshortening and shadowing. It is worth noting that $\gamma^\circ$ correlated with the AGB better than $\beta^\circ$ (beta-zero). Therefore, the $\gamma^\circ$ backscatter (VH or VV) was used to create the predictive models.

Coherence data are the second group of variables significantly correlated with the AGB. The correlation coefficients between the AGB and the coherence are negative, as the coherence decreases a with higher AGB. Figure 4 depicts the time series of the correlation coefficients between the AGB and VV and VH coherence during the year. The figure reveals that although both channels show a very similar evolution, the correlation coefficients between the AGB and the VV channels are higher than with the VH channels in most cases. According to [31], the response from the vegetated areas in the cross-polar channel at the C-band is expected to be more dependent on the vegetation layer than on the ground. As a consequence, VH polarization is more influenced by temporal decorrelation than VV. It was impossible to determine unequivocally the period when the relationship between AGB and coherence would be stable and significantly higher than in another period. However, the group with the highest correlations could be observed in the leaf-off period with snow cover (DOY 40 to 92), with the highest $r_{VV} = -0.68$ and $r_{HV} = -0.64$. Based on these findings, two prediction variables ($Coh_{VV\_avg}$ and $Coh_{VH\_avg}$) were derived for the AGB estimation by averaging the seven coherence pairs with the highest correlations from DOY 40 to 92. The correlation coefficient of the average coherence ($r_{VV} = -0.77$ and $r_{HV} = -0.72$) was higher than from the single pairs, and the average coherence was less affected by coherent noise. This confirms the importance of averaging coherent pairs.

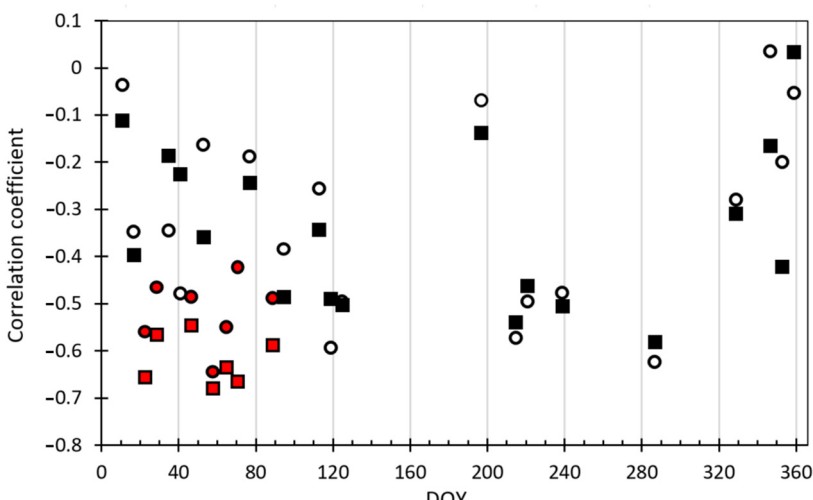

**Figure 4.** Correlation coefficients between AGB and the coherence of VH (circle) and VV (square) polarisation during the year. DOY, day of year. The coherence pair temporal baseline is 6 days combining S-1A and S-1B images. Red indicates the images used for the derivation of the average products. Correlations stronger than $-0.22$ ($-0.30$) are significant at the 0.05 (0.01) level.

In the case of the third group, Sentinel-2 DN values, the highest correlations with the AGB were observed for band B5 vegetation red edge ($r_{B5} = -0.85$) and band B4 red ($r_{B4} = -0.76$) from the top of the vegetation season (22 June 2017). In general, the red bands (B4 and B5) correlate higher than bands B8, near infrared (NIR), and B11, short-wave infrared (SWIR). For the analysed Sentinel-2 spectral bands, the relationship between the AGB and the DN values was negative. The exception is one significant positive case of B8 from the autumn season (Table 5).

**Table 5.** Correlation coefficients between AGB and Sentinel-2 bands from 4 periods in descending order.

| 28 January 2017 | | 29 March 2017 | | 22 June 2016 | | 30 September 2018 | |
|---|---|---|---|---|---|---|---|
| **Band** | **r** | **Band** | **r** | **Band** | **r** | **Band** | **r** |
| B5 | −0.65 *** | B4 | −0.65 *** | B5 | −0.85 *** | B4 | −0.74 *** |
| B4 | −0.64 *** | B11 | −0.65 *** | B4 | −0.76 *** | B5 | −0.63 *** |
| B8 | −0.57 *** | B5 | −0.62 *** | B11 | −0.47 *** | B11 | −0.56 *** |
| B11 | 0.07° | B8 | −0.49 *** | B8 | 0.01° | B8 | 0.44 +++ |

+++ highly significantly positive ($p < 0.001$); *** highly significantly negative ($p < 0.001$); ° not significant.

Considering the most correlated variables and removing the highly correlated predictors leads to a reduction in the number of models for evaluation. According to this rule, the variables correlating the highest with the AGB were selected from each predictor group. In total, five variables were included for further analysis: $B5_{22vi}$ (B5 band from 22 June 2016), $\gamma°_{VH\_leaf-off}$ and $\gamma°_{VV\_leaf-off}$ (average from ascending and descending images) and $Coh_{VH\_avg}$ and $Coh_{VV\_avg}$ (average of the seven highest correlations).

### 3.2. Performance of AGB Predictive Models

The MR1 and MPW models were created with the five predictor variables. The MR2 model is an extension of MR1 by including two additional terms, $Coh_{VH\_avg} * \times B5_{22v}$ and $\gamma°_{VV\_leaf-off}/B5_{22vi}$, as a common function of the predictive variables (Table 6). Their selection was based on the highest $R^2$ from all possible interactions derived by multiplying or dividing two predictive variables.

**Table 6.** Significance of the coefficient (predictors) in AGB models.

| Model | $B5_{22vi}$ | $\gamma°_{VH\_leaf-off}$ | $\gamma°_{VV\_leaf-off}$ | $Coh_{VH\_avg}$ | $Coh_{VV\_avg}$ | $Coh_{VH\_avg} \times B5_{22vi}$ | $\gamma°_{VV\_leaf-off}/B5_{22vi}$ |
|---|---|---|---|---|---|---|---|
| MR1 | ** | n.s. | +++ | ** | n.s. | n.a. | n.a. |
| MPW | *** | +++ | n.s. | n.s. | (*) | n.a. | n.a. |
| MR2 | *** | n.s. | n.s. | *** | n.s. | +++ | +++ |

*n.a.*, not applicable, and this predictor was not used in the respective model; *n.s.*, not significant; * varies around significance level $\alpha = 0.05$ during 500 bootstrap repetitions. +++ highly significantly positive ($p < 0.001$); *** highly significantly negative ($p < 0.001$); ** significantly negative ($p < 0.01$); * marginally significantly negative ($p < 0.05$).

The results (Figure 5) show that the MR1 predictive model estimated the AGB on AAL with the lowest accuracy ($R^2 = 0.78$ and RMSE 48.6 t·ha$^{-1}$). We tried to improve the performance of the MPW model by using the power Equation (6). Detected heteroscedasticity was eliminated by applying a logarithmic transformation to reach the residues' normal distribution (Equation (7)). To interpret the results, a back-transformation was performed (Equation (8)) with the coefficient of determination $R^2 = 0.80$ and RMSE 46.4 t·ha$^{-1}$. Bands B5 and $\gamma°_{VH}$ were found to be the most effective in estimating the AGB. The coherence VV polarisation channel contributed at a borderline 0.05 significance level. Other bands were not significant. This leaves open options for finding other explanatory variables and further improving the MPW model.

The most robust model (MR2), combining Sentinel-1 and Sentinel-2 bands, was derived from the predictive variables $B5_{22vi}$ and $Coh_{VH\_avg}$ and two additional terms, $Coh_{VH\_avg} \times B5_{22vi}$ and $\gamma°_{VV\_leaf-off}/B5_{22vi}$. The accuracy statistics reached RMSE = 41.2 t·ha$^{-1}$ ($R^2 = 0.84$). Although $Coh_{VH\_avg}$ was slightly more weakly correlated with AGB than $Coh_{VV\_avg}$, its influence in the predictive MR2 model was more significant. The presence of significant interactions indicates that the effect of one predictor variable on the response variable is different at different values of the other predictor variable. The effect is evident by the higher performance of the model and the lower RMSE reached by reducing the bias (Figures 5 and 6, Table 7).

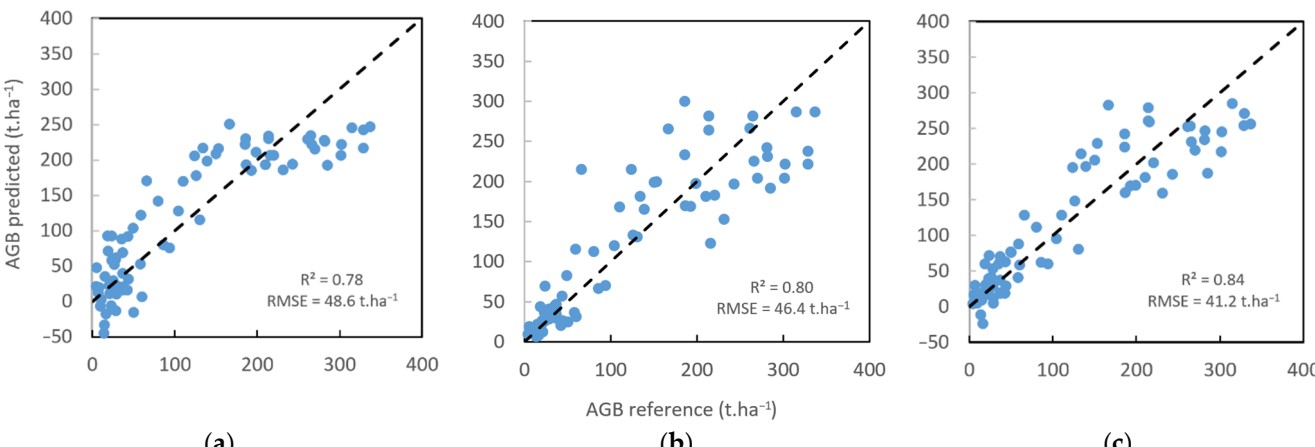

**Figure 5.** Reference versus predicted AGB for different regression models: (**a**) MR1, (**b**) MPW and (**c**) MR2. Predicted values were calculated using regression coefficients averaged from 500 bootstrap runs.

The correlation coefficients of the predictive models (MR1: $r = 0.88$; MPW: $r = 0.90$; MR2: $r = 0.92$) based on a combination of the Sentinel-2 band $B5_{22vi}$, the average $\gamma°$_backscatter from leaf-off without snow and the coherence from the leaf-off period from Sentinel-1 showed an improvement in the AGB estimates compared to the single predictive variables (Tables 4 and 5, Figure 4). B5 and coherence (VH or VV polarisation) were significant in all three models. The $\gamma°$ backscatter and the VH or VV polarisation significantly improved the AGB estimation in models MR1 and MPW. We consider that the term $\gamma°_{VV\_leaf\text{-}off} / B5_{22vi}$ better explains the relation to AGB than $\gamma°_{VV\_leaf\text{-}off}$, therefore the last one was not significant in the MR2 model. Further improvement in MR2 was reached by including the interaction term $Coh_{VH\_avg} \times B5_{22vi}$.

Dividing the reference AGB into three categories allows us to better understand and compare the performances of the MR1, MPW and MR2 models. The selected ranges approximately correspond with the AGB of shrubs (4–100 t·ha$^{-1}$), shrub–tree formations (100–200 t·ha$^{-1}$) and tree formations (200–350 t·ha$^{-1}$) and reflect the need to have a sufficient amount of reference data within each range. The assessment was based on quantifying the root mean square error (RMSE), bias and standard deviation of the error within each range (Figure 6, Table 7)

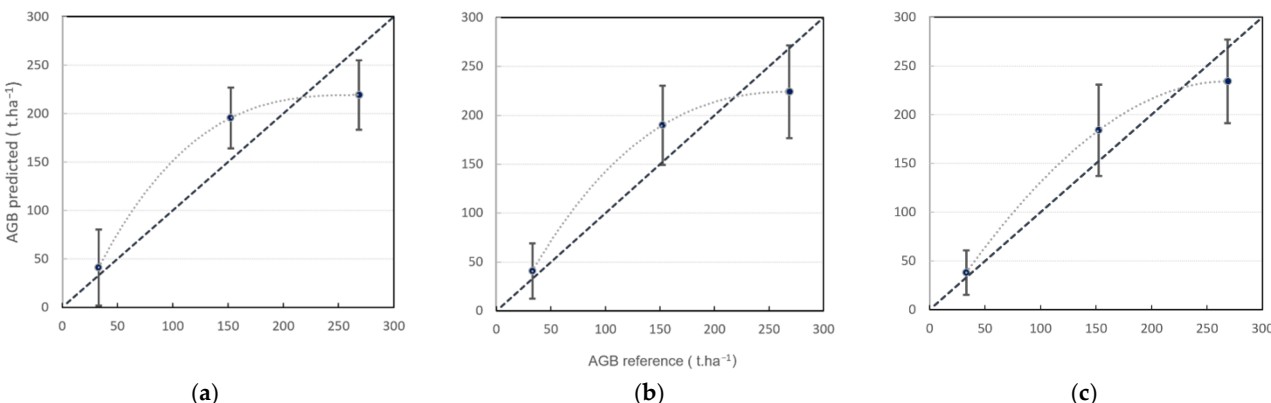

**Figure 6.** Comparison of the observed and predicted AGB averages per 3 reference AGB ranges for 3 models: (**a**) MR1, (**b**) MPW and (**c**) MR2. Error bars indicate a random error ($\sigma$ = SE) of the predicted AGB per reference AGB range. The dotted line indicates a fitting curve to calculated points (3rd order polynomial), and the dashed line corresponds to y = x line. If error bars do not overlap y = x line, bias is the dominant error in that AGB range.

The accuracy analysis reveals several commonalities in the predictive models (Table 7). The distribution of the RMSE, the RMSE%, the bias and the coefficient of variation (CV)

of the error across the reference AGB range is common to all three models. The models overestimate the AGB under ~200 t·ha$^{-1}$ and underestimate above this level. The most considerable overestimation bias occurs in the AGB middle range between 100 and 200 t·ha$^{-1}$ (Figure 6). This is not typical for a regression-based approach, where the regression curve passes through the point defined by the mean value of the reference and estimated data [16]. One explanation for this could be the distribution of the AGB in the sample plots with a lower representation in the range 100–200 t·ha$^{-1}$. In all models, the RMSE (in absolute units t·ha$^{-1}$) rises with a higher AGB. Relative RMSE and CV are the opposite, sharply decreasing with a higher AGB (Table 7).

However, there are remarkable differences in the balance between bias and random error in the RMSE. The bias error dominates in MR1 in the middle and highest range, while the random error dominates in the lowest AGB range. For the MPW and MR2 models, the random error is the dominant error source for all the ranges of AGB. It is also evident that the random error (SE) does not vary greatly across the different AGB ranges for all the models (Table 7, Figure 6).

**Table 7.** Accuracy of the models in terms of the averaged metrics from 500 bootstrap repetition stratified by the reference AGB range: sample size (N), root mean square error (RMSE), relative RMSE (RMSE%), bias, mean absolute deviation (MAD), standard deviation of the error (σ = SE) and coefficient of variation (CV) of the error (when CV > 1, random error dominates; when CV < 1, bias dominates).

| Model | Reference AGB (t·ha$^{-1}$) | N | RMSE (t·ha$^{-1}$) | RMSE% | BIAS (t·ha$^{-1}$) (MAD) | SE (t·ha$^{-1}$) | CV |
|---|---|---|---|---|---|---|---|
| MR1 | 0–100 | 42 | 40.0 | 121.2 | 8.2 | 39.2 | 4.8 |
| | 100–200 | 15 | 53.2 | 34.9 | 43.0 | 31.2 | 0.7 |
| | 200–350 | 20 | 61.1 | 22.8 | −49.4 | 36.0 | 0.7 |
| | Overall | 77 | 48.6 | 41.4 | 0 (42.4) | 48.6 | - |
| MPW | 0–100 | 42 | 29.3 | 88.8 | 7.7 | 28.3 | 3.7 |
| | 100–200 | 15 | 55.1 | 36.2 | 37.5 | 40.4 | 1.1 |
| | 200–350 | 20 | 64.9 | 24.2 | −44.3 | 47.4 | 1.1 |
| | Overall | 77 | 46.4 | 39.5 | 0 (32.9) | 46.4 | - |
| MR2 | 0–100 | 42 | 23.7 | 70.5 | 5.1 | 22.7 | 4.5 |
| | 100–200 | 15 | 56.4 | 37.0 | 31.6 | 46.7 | 1.5 |
| | 200–350 | 20 | 55.0 | 20.5 | −34.3 | 42.9 | 1.3 |
| | Overall | 77 | 41.2 | 35.1 | 0.3 (32.3) | 41.2 | - |

*3.3. AGB Estimation on AAL in the Study Area*

AAL was identified according to the methodology described in Section 2.3. The overall accuracy of the AAL map was verified on 127 randomly selected points and reached 89.8%, which confirms the potential for the operational deployment of the proposed method of AAL identification. In addition to 3214 hectares of forest land listed in the National Forestry Database [32] for the Viglas forest management units, we identified 992 hectares of shrub and tree formation on AAL. The total area covered by woody biomass is then 4206 hectares. Thus the proportion of AAL covered by woody biomass is 23.6% in the study area.

The AGB was estimated using the MR2 model for each pixel identified as AAL. The estimated AGB was divided into individual categories to clearly show its distribution on AAL. In determining the nine categories, we used the total range of AGB from 4 to 350 t·ha$^{-1}$ and established a simple interpretation of the legend, which we achieved by defining the step of 50 t·ha$^{-1}$. The MR2 results of AGB estimation grouped into nine classes are presented in Table 8 and Figure 7.

**Table 8.** Biomass estimation in Viglas according to MR2 model.

| Class | Area | | AGB | |
|---|---|---|---|---|
| t·ha$^{-1}$ | ha | % | Tonne (t) | % |
| <0 | 47 | 5 | 0 | 0 |
| 0–50 | 194 | 20 | 4858 | 4 |
| 50–100 | 221 | 22 | 16,590 | 14 |
| 100–150 | 183 | 18 | 22,913 | 19 |
| 150–200 | 145 | 15 | 25,410 | 21 |
| 200–250 | 104 | 10 | 23,352 | 19 |
| 250–300 | 60 | 6 | 16,401 | 13 |
| 300–350 | 26 | 3 | 8479 | 7 |
| 350+ | 12 | 1 | 4343 | 3 |
| | 992 | 100 | 122,346 | 100 |

Applying the MR2 model, parameterized for the range 4–350 t·ha$^{-1}$, the above-ground biomass on AAL was ~122,000 tons. The estimated AGB on forest land is ~750,000 tons according to the National Forestry Database for the Viglas FMU [32]. Total wood AGB is then ~872,000 tons in the whole area of the Viglas FMU. Thus the proportion of AGB on AAL is 14.0%, significantly less than the area proportion (23.6%). This indicates a lower mean AGB on AAL than on forest land. Indeed, the average AGB on AAL is 123 t·ha$^{-1}$, and on managed forest land it is 233 t·ha$^{-1}$. The obtained results for wood AGB on AAL according to the area of individual classes show an uneven representation (Table 8). The first three classes, with biomass from 0 to 150 t·ha$^{-1}$, represent up to 60% of the area. The occurrence of classes with AGB above 250 t·ha$^{-1}$ is only 10%. The predominance of low AGB classes indicates the dominance of shrub formations on AAL.

Abandoned agriculture land (AAL), agricultural land (AL) and their share according to altitude and altitudinal classes are depicted in Table 9. The three altitudinal classes were defined according to prevailing agricultural use associated with the slope of the terrain. The first altitudinal class (320–400 m a.s.l.) represents floodplains and slightly undulating hilly lands with compact urban and rural settlements and large-block arable land used for the production of feed cereals, potatoes and fodder. The third class (600–944 m a.s.l.) represents broken-up and inclined upland, traditionally exploited for agricultural in small plots of arable land, meadows, pastures and orchards, especially around the dispersed settlements. The second class (400–600 m a.s.l.) is a transition zone between the first and third class, with large-block arable land at lower altitude and extensive grasslands exploited for cattle and sheep breeding, and small plots of arable land at higher altitude.

**Table 9.** Occurrence of abandoned agricultural land (AAL) and agricultural land (AL) and AAL proportion on AL according to altitude.

| | 320–400 m | 400–600 m | 600–944 m | Overall |
|---|---|---|---|---|
| Total area of agricultural land (ha) | 2894 | 4587 | 996 | 8477 |
| Area of abandoned agricultural land (ha) | 162 | 554 | 276 | 992 |
| Share of AAL from AL (%) | 5.6 | 12.1 | 27.7 | 11.7 |

The spatial distribution of woody vegetation on AAL is uneven. The share of AAL increases with altitude and the distance from the populated part of the valley in the territory from 5.6 to 27.7% (Table 9). The abandonment process is evident in the second and third altitudinal classes (Figure 7).

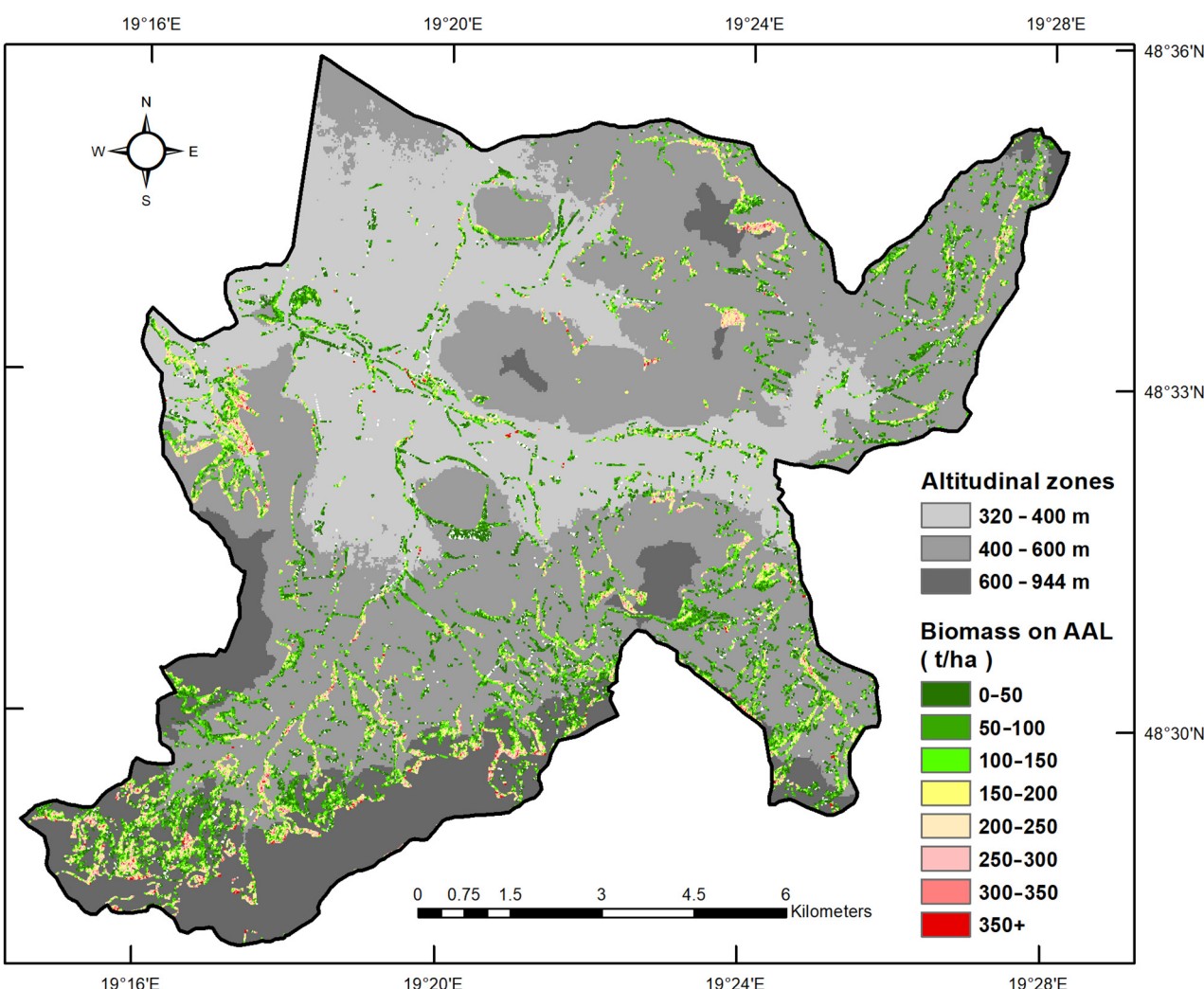

**Figure 7.** Distribution of predicted above-ground biomass on abandoned agricultural land in the study area of Viglas according to altitudinal zones (grey background colours). In order to enhance the visual perception, the number of pixels was magnified by a median filter with dimensions of 3 × 3 pixels.

## 4. Discussion

### 4.1. Remarks on the Proposed Approach of AGB Identification and Enumeration on AAL

The study's main goal was to propose and validate the concept of the satellite-based estimation of wood biomass on AAL. We designed the concept so that it could be practically implemented at the lowest possible cost. Therefore, when quantifying the biomass, we focused on the freely available Sentinel-1 and Sentinel-2 satellite data sources.

In identifying AAL, we used a combination of available cadastral data, LPIS and Sentinel-2 imagery. Although it is a practical and straightforward approach combining land use and land cover data, it is not used in national or large-scale AAL identification. The advantage of our approach is that, by combining layers, it eliminates the shortcomings of individual approaches. With the independent use of the LPIS layer, usually derived from aerial orthophotos, distinguishing small heterogeneous areas within land parcels is a problem due to the time-consuming task of interpreting images. The cadastre's legal status makes it possible to separate the wood vegetation growing on forest land and in urban areas, but it does not capture the actual land cover on agricultural land. By overlapping the plots outside the LPIS and cadastral data, we created a layer of potential AAL. Although verified only in the study area, it is expected that the proposed identification of potential AAL could be applied on the country level, and minimally on the European level, where

LPIS and cadastral data are available in a similar data structures. Specific to our approach, in the last step, we filtered out pixels under snow cover using Sentinel-2 imagery. In this way, we were able to mask arable land, meadow and pasture: thus, the remaining area represents real AAL with woody vegetation. It should be noted that the method must be modified in regions without snow cover.

Most of the earlier AGB studies suffered from difficulties related to the collection field data, resulting in inconsistencies between the field measurements and the AGB estimations [20]. A source of uncertainty is DBH and height values entering into allometric equations or not having a suitable allometric model for calculating the AGB [33]. Therefore, the essential prerequisite for AGB estimation modelling in our study was to prepare the high-quality reference data. We found from the field survey that wood vegetation on AAL creates heterogeneous structures formed by shrub, shrub–tree and tree formations. This heterogeneity required the use of specific AGB models for shrubs, young and mature stands and stumps. The volumes of mature trees, young trees and stumps were derived according to models parametrized for the Western Carpathian region [25–27]. Such models are missing or insufficiently documented for shrub vegetation [11]. For that reason, we derived the model from field measurements based on the mean height of *Prunus spinosa* as the dominant shrub species in the area of interest (Equation (1)). It could be argued that this allometric model will not work for other shrubby species. This may be critical if applying the model in areas where blackthorn is not the most represented bush species. We are aware of this critical limit, but this represents the first survey of shrub biomass quantification in Slovakia. There is no other way than to specify the AGB models for other shrub species gradually.

### 4.2. Procedures for Improving AGB Estimation

The remote sensing of AGB (using either reflectance or radar backscatter) is subject to decreasing sensitivity to AGB as the biomass increases. Besides saturation, other problems are speckle noise in the radar data and the heteroscedasticity of data, which mainly occurs when applying the AGB estimation power model. One study objective was to decrease the influence of the mentioned negative effects on the accuracy of AGB estimation. The added value of our research was testing and combining existing individual approaches [16–22,34–44], finding an optimal image processing method and selecting bands using multiple-linear and non-linear regression. The particular results are discussed below in three groups of predictive variables: radar backscatter, coherence and DN.

*Sentinel-1 radar backscatter:* Averaging and filtering the temporal intensity data is a simple yet effective approach to reducing speckle and temporal variations, as applied in other studies [34]. The seasonal variation of the relationship between the AGB and the radar backscatter was also shown in [35]. The authors calculated the correlation coefficients for 10 stacked and averaged images from 3-month periods in the range of 0.2 to 0.7. This indicates the importance of choosing an appropriate period for more precise determination of the AGB. In our approach, the database of 60 images was divided into three strata, corresponding to a leaf-off period with and without snow cover and a leaf-on period. The impact of multi-temporal image stack averaging and speckle filtering by the strata was demonstrated through Pearson's correlation coefficient between the $\gamma°$ radar backscatter and the AGB field values. The coefficient varied from 0.54 to 0.79. A comparison between the best-correlated strata and the whole sample revealed a positive effect of stratification. This was higher for VH polarisation, where the difference in the correlations was significant ($p < 0.01$) than for VV ($p = 0.08$). Further, we confirmed a higher correlation between $\gamma°$ and AGB in the leaf-off than the leaf-on period. The difference between leaf-off with and without snow cover was minimal and not statistically significant (Table 4).

Unexpectedly, $\gamma°$ and AGB correlations from the descending pass were higher than from the ascending pass (Table 5). We did not find a reason for this difference in our study or a reference to explain it.

A comparison of the correlation coefficients between the VH and VV polarisations did not reveal significant differences in general (Table 4). However, as stated in the previous paragraph, there were significant differences between the VH and (VV) polarisation variables themselves. Higher correlations were found out in the leaf-off period. This is in line with research completed by the authors in [35] related to VV polarization in the autumn–winter season. The opposite was found for VH, as our study showed the strongest, and the authors in [35] showed the weakest correlations in leaf-off periods. A possible explanation for the difference could be the AGB range from 100 to 400 t·ha$^{-1}$. In our case, shrubby wood vegetation up to 100 t·ha$^{-1}$ was also included. This suggests that the structure of the analysed stands may condition the variable selection for the AGB model.

As a predictor variable in biomass models, we used the arithmetic mean of $\gamma^\circ$ from averaged ascending–descending tracks, as they mostly eliminate the shadowing and foreshortening problem. Thus, areas of data distortion did not have to be masked out in our study.

Compared to other studies based on multi-temporal approaches to determining forest AGB with C-band, our findings indicate a closer relationship between $\gamma^\circ$ and AGB, with $R^2 = 0.59$ vs. 0.25 achieved from Sentinel-1 in [17]. Laurin [35] noted a max $R^2$ of ~0.49. A higher AGB accuracy ($R^2 = 0.73$) was achieved [36] through multi-date weighted averaging using C-band Envisat ASAR in tropical forests. The integration of our results from radar-backscatter analyses confirmed the assumption stated in objective three, that the averaged and filtered backscatter intensity from the properly selected period (leaf-off without snow cover) could improve the AGB estimation in the conditions of deciduous shrub stands in the Western Carpathians.

*Interferometric coherence:* Coherence images significantly correlate with the AGB. According to [22,36], a higher level of coherence is associated with winter observations and a lower level with summer/autumn observations. Our analyses revealed coherence variations during the entire year (Figure 4). A cluster of higher correlations was observed in the leaf-off period with snow cover (DOY 40–92), with the highest $r_{VV} = -0.68$ and $r_{HV} = -0.64$. Thus, our results confirm the seasonal behaviour of coherence reported in [22,36]. A possible elucidation of the variation in coherence between DOY 42 and 90 in our study is explained in [18,37–40] by variations of backscatter during freeze–thaw cycles. The authors in [38] compared observations under frozen and thawed conditions and found that the backscatter was lowest under frozen conditions when the trees and background were frozen. The highest backscatter was found for thawed and wet conditions because of the ice and water in the snow cover generating volume scattering. These variations in backscatter during freeze—thaw cycles, which occur on timescales of hours to days in the period of alternating plus and minus temperatures, are accompanied by large variations in temporal coherence [18]. Note that the coherence temporal baseline was 6 days. The distance between the two antenna positions and the respective height of ambiguity did not affect the absolute level of coherence in our study.

The coherence of the interferometric pair was more influenced by coherent noise than the averaged product, calculated from the seven highest correlations during the winter period with snow cover. Indeed, the correlation coefficient between AGB and average coherence ($r_{VV} = -0.77$ and $r_{HV} = -0.72$) was higher than from any coherence pairs (Figure 4). Therefore, similar to the backscatter conclusion, the multi-temporal average of the C-band Sentinel-1 coherence from the winter observations can improve the AGB estimation on AAL.

*Sentinel-2:* Our results confirm the suitability of Sentinel-2 B4, B5 and B11 bands for AGB estimation (Table 5). This set of bands was also well correlated with the AGB estimation of Scots pine from Sentinel-2 in [21], with the best $R^2 = 0.25$ for B12. Other band sets were found to be more robust: B8, B11, B12 or B5, B6, B8 in two research areas in [41]. Different sets of best S2 bands were identified depending on the regression method (LM, RF) [21]. As reported in [42], bands B11 and B6 were the most effective at estimating AGB in a Mediterranean forest ecosystem, with $R^2 = 0.46$ and 0.38, respectively. In our study,

we found that band B5 from the peak vegetation season showed a stronger relationship with AGB ($R^2$ = 0.72) than the above-mentioned studies. One possible explanation is that lower AGB does not saturate the signal, making the relationship between DN and AGB closer. Indeed, the AGB per hectare on AAL is markedly lower than in forest stands analysed in compared studies. Another possible reason for the better results is the more precise quantification of the AGB biomass on AAL training plots carried out in line with the recommendations for reducing the uncertainty in the AGB enumeration. This indicates that the robustness of spectral bands to estimate the AGB should always be analysed for each particular case.

Comparing the bands among the datasets from four vegetation seasons revealed the common feature that bands B5 and B4 (Table 5) had a higher correlation with AGB than B8 and B11 in all cases. The June dataset from the top of the vegetation season yielded the highest predictive accuracy for band B5 ($R^2$ = 0.72). The coefficient of determination from the other three datasets (January, March and September) for bands four and five varied in the range of 0.38 to 0.55. The reflectance in the red-edge spectral region (i.e., band B5) is sensitive to the green leaf area index (LAI), which varies in different phenological stages. Green LAI, defined as the total one-sided area of green leaves per ground area, represents a key variable in above-ground biomass estimations [43]. Therefore, it can be expected that a higher correlation at the top of the vegetation season (22 June) would be associated with the highest green LAI. In line with our conclusion, the top of the vegetation season (July) was shown to be the most suitable for predicting the growing stock volume of a Mediterranean forest from Sentinel-2 imagery [36].

*AGB predictive models:* Multiple regression is a powerful traditional statistical method often used in AGB estimation. According to the authors in [21], it is essential and worthwhile to examine its performance before using more advanced methods. We investigated three regression models linking the AGB with satellite observables (see Section 2.6). The MR1 multiple linear predictive model estimated the AGB on AAL with the lowest accuracy ($R^2$ = 0.78, RMSE 48.6 t·ha$^{-1}$), followed by the MPW model ($R^2$ = 0.80, RMSE 46.4 t·ha$^{-1}$). The MR2 multiple regression model performed the best, with $R^2$ = 0.84 and RMSE = 41.2 t·ha$^{-1}$ $\cong$ 62 m$^3$·ha$^{-1}$ (35%). The model is based on VH coherence (leaf-off), $\gamma°$ VV-backscatter (leaf-off without snow) and optical B5 band (leaf-on period) with two interactions between them: $\text{Coh}_{\text{VH\_avg}} \times \text{B5}_{22v}$ and $\gamma°_{\text{VV\_leaf-off}}/\text{B5}_{22vi}$. This clearly confirms our assumption defined in study objective four that an improved AGB estimation could be achieved by combining radar and optical satellite data [19,20,44] and multi-temporal radar backscatter and polarimetric coherence [17,22] and creating integrated models by multiple regression [45]. Concerning the components of RMSE, the random error is predominant, except in MR1, where the bias error occurs at AGB above 100 t·ha$^{-1}$ (see CV in Table 7). Both components of RMSE are a challenge for further reduction, e.g., by applying a post-processing bias reduction technique or, in the case of random error, using spatial averaging and more precise reference AGB assessment [32,46]. In line with [46], by defining homogeneous areas around 56 training areas, we have achieved (i) the positive effects of increasing the plot sizes on the predictive power of the AGB models due to speckle-noise reduction and positioning error reduction and (ii) a smaller RMSE% for larger plots compared to smaller plots. Further, by extension of the reference database by 21 tree plots by random sampling, ranging in their biomass from 100 to 350 t·ha$^{-1}$, we ensured (iii) a the similar number of training plots per the whole AGB range and fulfilling (iv) the even distribution of the plots throughout the territory to avoid pseudo-replication and to increase the generality of the training data. We are aware of possible inaccuracy at the delineation of the homogenous area around the plots. However, using available CIR and nDSM eliminates the problem and is the practical solution because the ground measurement of large shrub plots is complicated and time-consuming.

We can compare our results only with studies estimating forest AGB (or growing stock volume (GSV)) due to a lack of studies on shrub AGB. A study in the temperate forest of Poland [44] reached an RMSE of 60 t·ha$^{-1}$ (39%) and a saturation level around

200 t·ha$^{-1}$ based on fused Sentinel-1 and Sentinel-2 data and the random forest approach. The capability of the Sentinel-2 instrument to predict the AGB was exploited [41]. Applying an advanced k-NN method based on the random forest distance matrix in mixed deciduous broad-leaved forests in Italy, a notable RMSE of 27.1 m$^3$·ha$^{-1}$ (6.8%) was reached in Lazio and 41.7 m$^3$·ha$^{-1}$ (23.7%) in Tuscany. AGB estimation of tropical forest from Envisat ASAR C-band using a semi-empirical water cloud model (WCM) reached a remarkable R$^2$ of 0.9 and RMSE of 35.9 t·ha$^{-1}$ after multi-date weighted averaging [36]. Laurin [35] achieved good accuracy with AGB predictions in broad-leaved forests in central Italy (R$^2$ = 0.7, RMSE = 47 t·ha$^{-1}$ = 14%) by integrating Sentinel 1- and 2 and Sentinel-1 and ALOS-2 images. However, only 17 plots were analysed, with the AGB ranging from 100 to 400 t·ha$^{-1}$.

Studies based on the L and P bands generally reached better results than those using a single channel; for example, RMSE of 32.2 m$^3$·ha$^{-1}$ (34%) was reported in [22] in boreal forest with L-band ALOS PALSAR when a multi-temporal HHVV coherence was combined with a multi-temporal HV-backscatter. A promising RMSE of 30.1 t·ha$^{-1}$ (20.8%) was reported [47] in a hemi-boreal forest based on airborne SAR data combining L and P bands with forest height derived from PolInSAR. A comprehensive comparison of the approaches to forest biomass retrieval from EO in different biomes was published in [16], with RMSE% varying from 37 to 67%. The authors concluded that all current spaceborne sensors (SAR and optical) are inadequate for accurately estimating AGB beyond 100–150 t·ha$^{-1}$, and for a concave curve, such as the kind produced by saturation, it is inevitable that overestimation will occur for low biomass and underestimation will occur for high biomass if linear regression is applied.

By including the additional terms into the regression model MR2, we achieved a shift in the saturation point to ~230 t·ha$^{-1}$ and an accuracy (R$^2$ = 0.84, RMSE = 35%) comparable to more advanced methods applied in studies quantifying forest AGB. We emphasise that our results are related to complex shrub, shrub–tree and tree formations, and may be a benchmark for further studies of AGB estimation on AAL. As far as we know, this is the first such study in a temperate climate zone of the Western Carpathians to include the derivation of an allometric model for blackthorn (*Prunus spinosa* L.) in this region. Deriving the model was a prerequisite to creating a consistent reference database for the whole examined AGB range from 4 to 350 t·ha$^{-1}$. We are aware of the limitations of the model, since it was built for blackthorn. As it is the most abundant shrubby species in the study area, we assume that the reference database is sufficiently reliable. Note that our AGB estimation took into account the whole shrub or tree biomass with bark and stump. The biomass of leaves was not included due to missing models determining the leaf biomass of shrubs, or in the case of trees, such models are available only for some tree species.

### 4.3. Economic and Environmental Aspects of Agricultural Land Overgrowth

Our results show that the proportion of AAL covered by shrub and tree formations (992 hectares) among the total area covered by woody biomass (4206 hectares) is 23.6% in the Viglas study area. According to the National Forest Inventory, the percentage of agricultural land covered by tree biomass is 13% of forested land in Slovakia [48]. However, it is impossible to compare these two percentages, as the share of AAL1 and AAL2 classes in the study area is unknown.

In terms of AAL occurrence on agricultural land (AL), it depends on the altitude and related population density. The total share of AAL with woody biomass on AL is 11.7% in the study area and ~18% nationally [5]. Thus, the proportion in the study area is still lower than the national average. However, this share increases with altitude and the distance from concentrated settlements in the valley from 5.6 to 27.7%. The abandonment process is evident in the southwestern, southeastern and northeastern parts of the region in mountainous and remote areas with less productive soils in the second and third altitudinal classes. The highest share of AAL on AL is 27.7% in the third altitudinal class (see Table 9 and Figure 7).

The area of AGB classes shows an uneven representation (Table 8). The first three classes represent up to 60% of the area, with AGB from 0 to 150 t·ha$^{-1}$, whereas only 10% of the classes represent AGB over 250 t·ha$^{-1}$. The predominance of classes with low AGB indicates the dominance of the shrub formations on AAL in the first stages of overgrowth. This is also confirmed by the more or less homogeneous composition of shrubs dominated by blackthorn (*Prunus spinosa)* in our 56 training plots. Similarly, findings in a comparable Budzów region in the Polish Carpathians [9] showed that the abundance of species that encroach on unmanaged arable land and pastures creates a composition of various plant heights, sizes and distributions that is more uniform during the early stages and more varied as time progresses.

In addition to the overgrowing of meadows and pastures, according to [5], historical landscape elements and mosaics are gradually disappearing under the influence of ongoing secondary succession on a national level. These are narrow-band fields, overgrown ramparts of collected stone and landscapes with scattered settlements, which represent the area's high potential, with extraordinary cultural and natural value. These typical elements in our study area, Viglas, have been replaced with shrub and tree formations, especially in the third altitudinal zone over 600 m. Summarising the acquired knowledge about the AGB, and its spatial distribution and shrub species composition, we can state that the situation of overgrowing areas is not satisfactory from either an economic or environmental point of view.

## 5. Conclusions

In this study, based on the 11.7% share of AAL acreage on agricultural land and an average AGB per hectare significantly lower than on forest lands (123 t·ha$^{-1}$ vs. 233 t·ha$^{-1}$), we proved that the abandonment of cultivated agricultural land is a phenomenon that is emerging as a severe economic and ecological landscape problem.

Without precise data about AGB and its area and increment on AAL, it is impossible to set up a sustainable system for its economic use or assess the negative and positive effects of land overgrowth regarding biodiversity. Therefore, the problem of agricultural land overgrowth is still an extremely topical issue. Based on our findings in the Viglas model area, we propose to focus further satellite-based research on the following: (i) refining the identification of woody vegetation on AAL, (ii) defining spatial units with similar overgrowth structure as a basis for economic and environmental planning, (iii) further improving the models for quantifying wood biomass to a level of accuracy that allows changes in AGB and its increments to be evaluated in shorter than 10-year cycles and (iv) setting up an AAL management system. As our study was limited to Sentinel data, we see potential to improve the AGB estimation on AAL by applying multi-frequency acquisitions with L and P bands, full polarimetry and the further development of PolInSAR and radar tomography methods.

For the operational deployment of the proposed concept of AGB estimation on AAL, we also see the potential of using technology such as Google Earth Engine (GEE), which allows work with RS data without downloading them and offers appropriate computing power to classify imagery.

**Author Contributions:** Conceptualization, T.B. and J.F.; Methodology, T.B., J.P. (Juraj Papčo) and M.S.; Field campaign design, J.P. (Jozef Pajtík) and I.S.; Resources, J.P. (Juraj Papčo) and J.P. (Jozef Pajtík); Investigation, T.B., I.B. and I.S.; Data curation, J.P. (Jozef Pajtík) and M.S.; Writing—original draft preparation, T.B.; Writing—review and editing, T.B.; Project administration, T.B.; Funding acquisition, T.B. and J.F. All authors have read and agreed to the published version of the manuscript.

**Funding:** This research was funded by the Government of Slovakia through ESA contract no. 4000123812/18/NL/SC under the Plan for European Cooperating States. The views expressed herein can in no way be taken to reflect the official opinion of the European Space Agency. This research was supported by the Slovak Research and Development Agency in the framework of the project "Development of advanced geospatial technologies for multiscale monitoring of forest ecosystems" (APVV-19-0257). The paper was also supported by the Slovak Scientific Grant Agency (VEGA)

under grant no. 2/0023/19, Land cover dynamics as an indicator of changes in the landscape. This publication is the result of the project implementation Centre of Excellence of Forest-based Industry, ITMS: 313011S735, supported by the Research and Development Operational Programme funded by the ERDF.

**Conflicts of Interest:** The authors declare no conflict of interest.

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
