# Peer review of "Woody Above-Ground Biomass Estimation on Abandoned Agriculture Land Using Sentinel-1 and Sentinel-2 Data"

_remotesensing, doi:10.3390/rs13132488_

Round 1
Reviewer 1 Report
The manuscript titled “Shrub and Tree Biomass Estimation on Abandoned Agriculture Land Using Sentinel-1 and Sentinel-2 Data” presents describes the testing of optical and SAR satellite-based modeling of above-ground biomass in abandoned agricultural lands in Slovakia. I believe the paper requires a significant level of improvement before it is acceptable or appropriate for publication.
- I strongly suggest there be a significant amount of English editing – the numerous grammatical errors and awkward sentences render the content confusing and sometimes very unclear.
- The title states “shrub and tree biomass”, and though the authors are estimating both of these, there is not comparison or analysis of tree versus shrub biomass, which would be nice to see. I would suggest changing this in the title to “woody above-ground biomass” to better reflect the paper.
- There are a few places where acronyms are not defined the first time they are used (e.g., ha, nDSM, NIR, SWIR).
- To me, the introduction does not flow very well or really explain the “why” behind the study very well – why is estimating this particular AGB so important? It would be better to first explain the AAL and reforestation, then why it’s important to estimate AGB in these areas, what methods exist for doing so, and what has been done so far. Then, state the knowledge gap that is being filled by the work, and the objectives. Also, the paragraph on lines 87-91 does not actually state a hypothesis, but rather an objective. I would suggest that the authors state objectives rather than listing hypoetheses. I’m also not very clear on the main differences between the first and third hypotheses/objectives – it seems that the first is just a smaller piece of what is described for the third.
- The methods section is not clear enough or detailed enough in some places. For instance, biomass is in units of weight per area (g/m2 or tonnes/hectare, as the authors use). However, in numerous places (in methods and elsewhere in the paper), the authors refer to volume measurements and stock measurements or estimates. I do not have a strong background in biomass measurement or estimation, and it seems that the authors will have converted measured volume into biomass. The authors do not state this explicitly or explain it very clearly, however, and do not make their definitions of and differences between volume, stock, and biomass clear. Some of the further places needing clarification are found in more specific comments below.
- The authors define two AAL classes (shrub vs treed), but do not use this division in their methods or results, from what I can observe, and these classes are not mentioned again until much later in the paper. Where and how these classes were used needs to be made explicit, and included in the results. Or, if they did not play a role in the authors’ methods, they should not be part of the paper.
- There needs to be a small section describing how AAL areas were identified/mapped. This is described in another paper (which is not currently available, and so I have no way of evaluating it or truly understanding it). But it should be described here as well, with more than just one sentence.
- The results should contain some information on the accuracy of AAL location maps, and some assessment if possible on the quality of the AGB estimates from allometric equations.
- The authors often list/describe/discuss S1 backscatter, S2 spectral bands, and then S1 coherence, in that order. I find this awkward. As the coherence is a radar product I would strongly suggest it be placed before the S2 variables/inputs when discussed/described.
- Was there any significance testing done to determine whether the MR2 model performed significantly better than the ANL or MR1 models? The R-squared values for the MR2 and ANL are similar, and there may not be a statistically significant difference.
- The discussion section contains a lot of repetition and paraphrasing of content already provided in the methods second, and not a lot of actual discussion. The authors should revisit each of their objectives in here, explicitly discuss how their objectives area met, and also dig deeper into what their results mean (e.g., why were the chosen input variables the strongest predictors? What do the two additional terms that account for variable interaction say about the remote sensing of AGB and why are they the most important ones?), and what the implications are for further above-ground biomass estimations in abandoned agricultural areas.
*** Specific Comments ***
Ln 30: page numbers only need to be given in citations when a direct quotation is used.
Ln 40-42: please explain this better. What is the registered land use type of these grasslands, and what is their actual use?
Ln 47: what do you mean by “the height of succession formations”?
Ln 63: [16] concluded that current satellite sensors aren’t adequate for accurately estimating biomass beyond 100/150 t/ha; so how do the authors know their work is different? What makes it different? How are they addressing the saturation problem? This isn’t very explicitly addressed.
Ln 84: I disagree with using the term “fusion” here (and elsewhere)…. Inputs from S1 and S2 are combined in the modeling, yes, but they are not really fused together as I generally understand the term from a remote sensing point of view. The authors also seem to suggest here that combining optical and radar is unique and novel, but according to their citation [16] it is not. Their use of Sentinel may be novel, however. If that is a key contribution, then that should be emphasized.
Ln 112: please describe the National Cadastre and LPIS – what do they comprise? What date(s) do they represent? How were they compiled?
Ln 126: it should be “maximum” not “max”
Ln 152: how did you assess the accuracy of canopy closure estimates on the plots, to get this 5%? How was upper height calculated (and how is it defined)? Does “canopy closure” only refer to shrubs? And are “shrubs” identified by species, or by species and height?
Ln 157: is it safe to assume that this allometric model worked as well for other shrubby species? Is there any information/data to back that up?
Ln 162: there needs to be an explanation of the nDSM layer somewhere.
Ln 162-166: where these homogeneous areas around the plots a unit of analysis? How exactly did they factor into the authors’ methods? How were they used in subsequent modeling? Was the modeling itself based on per-pixel values, or was it based on per-object information? If the latter, the varying sizes of these homogeneous objects would have important implications for the modeling and its results and then its application. This needs to be addressed.
Section 2.3.1: How were mature trees vs stumps vs young trees identified/defined? There needs to be more explanation of how biomass was estimated in here. I would recommend brief descriptions rather than just referring to another paper.
Ln 202: how many images covered this period?
Ln 206-7: what is “1sek”? What is “bisnic 5-point interpolation”?
Ln 217: “table” should be “Table”, here and elsewhere in the paper; also, “figure” should be “Figure”, too, when referring to a specific table or figure.
Ln 219-220: the wavelengths of the specific bands should be given, and DN should be defined.
Ln 225: by “individual plots” do you mean ground plots (which are of various sizes) or the homogeneous areas (which are also of various sizes)? How do you deal with the issue of various sizes – e.g., a larger plot will contain more spectral variability simply because it has more pixels?
Ln 231-234: I am not clear on what makes the ANL model an “allometric” equation, and different from the multi-variate regression models. Please explain.
Ln 249-254: this is a long quotation; I would avoid direct quotations unless necessary, and paraphrase/rephrase from the original source instead
Ln 277: what’s “Sigma-zero”?
Table 5: there is no explanation as to what the numbers associated with each S2 band variable mean – e.g., what does “22vi” or “29iii” mean?? These need explanation.
Ln 297: because correlation coefficients are negative, the “Correlations above…” does not make sense. “Correlations stronger than….” or “…below -0.22 (-0.30)” would make more sense.
Ln 300: I think something like “variable type” is better than “predictor group” – I don’t think the latter is a very clear term.
Ln 305: there needs to be an explanation in here on how you selected/identified the two additional terms, and why.
Ln 330: what do you mean by “… improved AGB estimation in all three models”? Improved compared to what?
Figure 6: I think the purpose of this needs to be better explained. E.g., what are the three reference AGB ranges, and why are they important? This needs to be described in the text and addressed more directly and clearly.
Table 7: again, please explain in the paper first why you are dividing up results based on these ranges, and how/why you chose those ranges. Same comment for Table 8.
Ln 352: where does the information on amount/location of forest land come from?
Ln 356-357: what is “ths.”? Does it mean thousand? If so, just put 122,000 tons instead.
Ln 359: “FMU” is not defined.
Ln 362: saying “results on wood stocks…” is misleading to me. AGB is not the same as wood stocks. If you want to discuss stocks, then explain what you mean by this if it is different from AGB. If it isn’t, then use AGB.
Table 9/Ln 368: what do you mean by “occurrence of AAL on AL”? I think you mean the proportion of total agricultural land that is AAL? Please explain this. Also, please explain why you are listing these numbers divided into elevational groups before discussing/presenting them – what is the importance/why? I think there are some interesting patterns here. You just need to explain the reasoning first.
Ln 447: I don’t believe growing stock volume and AGB are the same thing. For one thing, AGB is weight, and volume is volume. They also represent two different ways of measuring woody vegetation.
Ln 453-457: Can you speculate on why your results (R-squared) are so much stronger than results from other studies? What are you doing differently that could contribute to this?
Ln 475: I don’t understand what you mean by “the whole volume of the forest is subject to change in the day cycle, especially in the characteristic period of alternation of plus and minus”. What do you mean by “volume of the forest”? Do you mean volume scattering? And what do you mean by “alternation of plus and minus”? Are you meaning in temperatures?
Ln 552: again, “AGB mean volume per hectare” makes no sense – AGB is not volume. Please make sure you use the correct terms for the correct concepts. Also, you use “ha” instead of “hectare” throughout most of the paper. Be consistent in using the same term everywhere.
Reviewer 2 Report
This manuscript describes a methodological paper describing a new way to estimate shrub above ground biomass from satellite data in abandoned farms. The topic fits well with the journal´s aim and scope, and it may be of interest for the journal´s international readership. The mansucript is in general well written and the the methods carefully presented and well applied. The results are well described. The weakest part of the mansucript is the discussion, with a few points that the authors could consider to improve:
- Although the introduction provides clear initial hypothesis, there are no comments in the discussion if such hypothesis are to be accepted or rejected.
- The clear over- and understimation of AGB are not substantially explained, neither the consequences of such missestimations for the estimaiton of the AGB in the target area.
- The discussion is bascially statistical and methodologicla. There are very little effort to explain the ecological mechanisms behind the observed results, and hence I don´t think the section 4.3 is supported by the results or the rest of the discussion.
All things consiered, I recommend accepting the manuscript for publication after such changes are done.
Reviewer 3 Report
The title of the manuscript (MS) deals with "Shrub and Tree Biomass Estimation on Abandoned Agriculture Land Using Sentinel-1 and Sentinel-2 Data". The topic of this manuscript is of interest and well written and I liked reading it, great job!
Just a few comments.
In the Keywords: Please remove Abandoned agricultural land, no need to repeat it since it is already mentioned in the title
In the "Introduction" section, the authors should provide several references about the use of Remote Sensing (RS) technology for estimating the shrub and tree biomass, and how the modern RS technology such as the GEE platform that enables scientists and researchers to analyze real-time changes to the Earth’s surface, for example:
https://www.nature.com/articles/s41598-020-69743-z
https://link.springer.com/article/10.1007/s41976-019-00020-y
https://www.sciencedirect.com/science/article/pii/S146290110700024X
https://iopscience.iop.org/article/10.1088/1748-9326/2/4/045025/meta
In the "Discussions" section, the authors should talk about the applicability of this method to another case study abroad. What are the mandatory prerequisites? Is it applicable worldwide?
Round 2
Reviewer 1 Report
The revised manuscript “Woody Above-Ground Biomass Estimation on Abandoned Agriculture Land Using Sentinel-1 and Sentinel-2 Data” is greatly improved from its previous version, and I appreciate all the hard work the authors put into it. The document as a whole is much clearer and more straightforward in it structure. The discussion section is also very much improved. Thank you.
Here are my comments on the revised version of the manuscript:
Ln 42: the sentence “The process of overgrowth…” is a bit confusing in its wording. Perhaps something like “Areas of shrub and tree overgrowth are largely labeled as “permanent grasslands”… “ would be better?
Ln 51: I would use “comparing land cover from…” rather than “the situation” – the latter is a very vague term.
Ln 80: I would use “Besides saturation, another problem is…”
Ln 89: specify “This study” or “Our study” here
Ln 100: I think you need to explicitly explain here why you chose S-1 and S-2 inputs (especially given that L or P band radar performs better in AGB estimation).
Ln 117-119: as you use a comma in “12,518” I would suggest considering using a consistent format for any numbers >999., E.g., “3,214” and “8,477”. But this may be more of a personal style.
Ln 163: it would be nice to clarify/explicitly state that these 56 plots ranged “in their biomass from…”, rather than relying on the units to convey this information. This same comment can apply elsewhere in the manuscript, since portions of the analysis are categorized by biomass level for some steps, and by elevational gradient or by type (tree vs shrub) in other steps. Being explicit helps the reader to know to what exactly what you are referring.
Ln 187-198: it is confusing to me to have this description of creating vector polygons for linking ground biomass estimates to satellite layer statistics in with shrub biomass estimations, especially since it falls between the allometric equation development. I think this should probably be its own small section, or at the beginning of the ‘Statistical models’ section. Another question: if this is described in the shrub biomass section, is this method used only for shrub plots, or also for treed plots? Please state this explicitly in the text. I also would like to see you address the issue of variable homogeneous polygon sizes (e.g., in the Discussion) – how do these variable sizes influence the modeling results? Is it reasonable to assume constant AGB across those homogeneous areas?
Ln 262: “pixel resolution” is better than “spacing”
Ln 267: please explain in the text why modeling was based on per-object information, while the prediction models were then applied at the pixel level. You addressed my previous questions on this very thoroughly in your reply (thank you), but I think more explanation also needs to be put into the paper itself. Not a long explanation, but an explanation of some kind. Also, since these are two different units of analysis, and likely has some implications for the results, addressing this would be a good addition to the Discussion.
Ln 272: there needs to be a better explanation in the text here to explain what you mean by “two additional terms as a common function of predictive variables.” You provide a nice explanation in your response to my comments (thank you again), but this is still unclear to me in the manuscript. Perhaps add a sentence or two explaining why and what you did: e.g., that you tested additional terms that account for interaction between any two predictor variables (e.g., one variable divided or multiplied by another), using R^2 to select terms that were most beneficial to the model’s explanatory power. I.e., the explanations given in Ln 349-352 should be up here I think.
Ln 273-274: I’m still not sure I agree with the use of “allometric” to describe your third model, even though it follows the general structure of an allometric equation (i.e., regression plus a scaling term). To me, the term “allometric” refers to biological traits/variables, but your equation is based on spectral properties. If you wish to use this term, perhaps make it explicit that the equation itself doesn’t have biological variables in it, and this refers to the structure of the regression itself instead.
Ln 298: should be “relationships” rather than “relations”
Ln 348: Title for section 3.2 should be “models” not just “model”
Ln 361: “channel” rather than “cannel”
Ln: 409-447: I would put this into its own section, since it is applying the best model to predict AGB the study area, and then analyzing the distribution of both biomass-based and elevational classes or categories across the area. I.e., this is not part of assessing model performance in my opinion and should be separate.
Ln 466: Thank you for adding this detail. However, a brief explanation of how this was done (e.g., like the one you gave in your response to reviewers) belongs in the Methods, and this piece of information belongs in Results. This is an important part of the work – it is a foundational step in the work.
Ln 487: there are some reference source errors here and in the following paragraphs.
Ln 485: perhaps “The study’s overall goal was to…”, unless this is referring more to one of your specific objectives. Then it should be “One study objective…”
Ln 506: I’m confused… Does this mean there were no significant differences between VH and VV polarizations in general, but that there were significant differences between VH polarization variables themselves (as stated in previous paragraph)? Please make this clearer in the text.
Ln 519: The term “synthesizing our results” is very vague to me. I think you mean, integrating your three groups of predictor variables? Please be clearer here.
Ln 608: is it normal practice to not include leaves in AGB estimations? Please explain in the text why this was the case, and discuss the implications for your results.
Author Response
We wish to thank the reviewer for constructive comments in this second round of review. The comments provided valuable insights to refine the article contents. Especially we thanks for very concrete suggestions on how to correct various ambiguity and inaccuracy. We have taken all comments into account to improve and clarify the manuscript.
